# Fertility Decision-Making in the UK: Insights from a Qualitative Study among British Men and Women

**Mikaela Brough** [1,*] and **Paula Sheppard** [2]

1   School of Engineering, Physical, and Mathematical Science, Royal Holloway University of London, Egham TW20 0EX, UK
2   School of Anthropology and Museum Ethnography, University of Oxford, Oxford OX2 6PE, UK
*   Correspondence: mikaela.brough.2022@live.rhul.ac.uk

**Abstract:** Scholars are interested in better understanding the low fertility observed in higher income countries. While some people are choosing to have smaller families, countries also report a 'fertility gap', which is the proportion of people who end up with fewer children than originally desired. This paper investigates some causes of the fertility gap in the UK. We amassed qualitative data from seven focus groups conducted among men and women of reproductive age with different educational backgrounds. These focus groups suggest that social support is an influential factor for Britons thinking about having children, although discussions differed in terms of whether this was support from partners or parents. Discussions with university-educated women featured themes of career opportunity costs, and non-university men contributed insights on the financial burden of parenthood. This exploratory study provides up-to-date material on unwanted childlessness and the low fertility in the UK, and highlights the merit of using qualitative methods in understanding the fertility gap.

**Keywords:** reproductive decision making; fertility gap; qualitative study; United Kingdom; low fertility; focus group study; education and gender; anthropological demography

## 1. Introduction

Across Europe, fertility rates show a downward trend albeit with much variation between countries. The decline in the period total fertility rate below the 'replacement level' (2.05 children per woman) represents a transition in Europe towards generally low fertility (Sobotka et al. 2018). This transition implies dramatic repercussions on the social and economic fabric of Europe, specifically regarding labour markets, gender relations, public policy, and family configurations. Scholars (Basten et al. 2014) have identified a variety of causes for this change, some of which include expanding education for women, rising income, increased gender equality, economic uncertainty, globalization, family dissolution, and expansion of sexual education. Many other complementary and complex forces have been outlined in the scholarly effort to understand declining fertility rates.

While some low fertility is by choice (Berrington 2017), that is, people wanting fewer children, a portion of this decrease is involuntary (Payne et al. 2019). Beaujouan and Berghammer (2019) compared the fertility intentions of 20–24-year-old women born between 1965–1979 to their outcomes at age 40 in the United Kingdom and determined this cohort had on average 0.3 fewer children than the number they said they desired. This disparity between fertility intentions and outcomes is also known as the 'fertility gap' (Chesnais 1999; Goldstein et al. 2003). Demographic research has determined one important factor driving an overall declining fertility rate and the fertility gap in the UK is postponement, meaning that the average age of first birth is ever increasing. In 2016, the mean age at first birth approached 28–30 years old in low fertility countries, which is a marked difference from 1970, in which most women gave birth before age 25

(Beaujouan 2020; Sobotka et al. 2018). While postponement does not always perfectly correlate with lower fertility in some countries and contexts (Beaujouan and Toulemon 2021; Beaujouan 2020), a common consequence of postponement is a shortened window of viable reproductive time, which leads to families possibly having fewer children than desired (creating the gap). Since postponement has become an influential fertility pattern of the last century, understanding this pattern, and the corresponding low fertility rate and gap is a primary concern of demographers, economists, and social scientists (Ajzen and Klobas 2013; Goldstein et al. 2003; Philipov 2009).

This article aims to identify the most important self-defined barriers to reproduction for men and women resident in the UK using qualitative methodology. This small-scale study is part of a larger pilot mixed-methods project. According to Clavia T. Williams-McBean (2019), conducting a qualitative phase in a mixed methods pilot study is useful for six main reasons: to develop research instruments, assess the feasibility of recruitment protocols, assess research protocols, collect preliminary data, pre-empt possible challenges, and increase researcher confidence. In the case of this study, qualitative methods were chosen as part of a larger mixed methods pilot to help the researchers gather preliminary data and refine research instruments, specifically to give an initial idea of what is important to people to help design other portions of the pilot.

A series of online focus groups were conducted, separately for men and for women, and to assess what differences there may be across socioeconomic strata, these were conducted separately for more and less educated groups as well. Demographic research has shown repeatedly that participation in higher education (particularly for women), is linked to fertility postponement in Europe and the United Kingdom (Sobotka 2004). With highly educated people entering the workforce at an older age, scholars have theorized that this shifts the standard milestones of the life course (meeting a partner, getting married, having children), to an older age. In addition, education is one of the main three markers of socio-economic position (SEP), more generally linked to postponement (Singh-Manoux et al. 2002). In this study, 'completed a university degree' was used as a cut-off for dividing participants into more and less educated groups. While having a bachelor's degree is an imperfect predictor of social position, it proved to be a simple and effective metric capable of generating diverse discussions. The researchers used conventional thematic analysis to draw out the main ideas in the data, again focusing on differences in gender and education.

### 1.1. From Macro to Micro Perspectives

At present, much of the research in this field has aimed to quantitatively identify the social, economic, and demographic variables responsible for influencing fertility patterns. Results from this research have been sweeping, providing information (McAllister et al. 2016) pointing to variables such as education level (Balbo et al. 2013; Ní Bhrolcháin and Beaujouan 2012; Huber et al. 2010), family background (Kramarz et al. 2021; Tropf and Mandemakers 2017), and wealth (Colleran and Snopkowski 2018; Fieder et al. 2011) in terms of their influence on relevant metrics such as Total Fertility Rate (TFR) and age of first birth. For instance, scholars have determined that international timings of first births can be somewhat attributed to differences in age of educational completion and consequent age of entry into the labour market, with high levels of education being correlated with lower fertility outcomes (Nicoletti and Tanturri 2008). Beaujouan and Berghammer (2019) confirmed that highly educated women in Europe have a lower mean number of children (except in Belgium and Norway) and a higher level of childlessness (except in the Czech Republic and Norway). In the UK, they observed a small fertility gap among lesser educated women, but a marked gap among highly educated women. Compared to other European countries, the UK displays a moderate gap in expected fertility vs. completed fertility and is most like Lithuania, Germany, and Hungary in terms of the difference between actual vs. intended fertility (~0.3) (Beaujouan and Berghammer 2019). The UK (as well as the US) also has shown strong educational differences in terms of fertility when compared to other European countries, driven by higher rates of unplanned births among lower educated

groups (Berrington et al. 2015). Current data on the UK fertility gap only compares the intended vs. completed fertility of women, not considering the intended family sizes of men. The existence of certain national family policies (e.g., family allowance, maternity leave) has also been shown to have an impact on a country's relevant population metrics (e.g., TFR) (Luci-Greulich and Thévenon 2013).

Despite the rich abundance of macro-level quantitative work on factors of low fertility (including other relevant metrics such as postponement, fertility gap, etc.), taking complex individual factors into play is another crucial concern in the literature. For instance, Tropf and Mandemakers (2017) claim that explanations put forth to explain the correlation between participation in higher education and fertility postponement can sometimes be spurious, with unobservable background individual factors such as personal upbringing influencing the relationship. In addition, the results of quantitative research on the causes of low fertility are highly contingent on the national/institutional context (Ajzen and Klobas 2013). For instance, Del Boca et al. (2009) found that in Europe, social policy systems can be grouped broadly into four main types—pro-family (e.g., France and Belgium), pro-traditional (e.g., Italy and Spain), pro-egalitarian (e.g., Denmark and the Netherlands), and non-interventionist (e.g., the UK). They write that the UK is considered a non-interventionist state because it has an underdeveloped system of social welfare consistent with an ideology of governmental non-interference. Del Boca et al. (2009) found differences between these four types of countries in terms of fertility outcomes, with women in pro-traditional countries having the highest fertility across cohorts. In comparison, lower educated women in the UK had lower fertility outcomes, which the authors attribute to the "income effect;" for women with lower education and a lower potential income, having one child negatively affects the probability of intending for another due to the high costs associated with childrearing, which is more pronounced in a non-interventionist country. Overall, this matrix of education, institutional policy, and individual background points to the holistic nature of this issue, in which micro, meso, and macro level considerations interact.

Beaujouan and Toulemon (2021) claim that the fertility gap is driven by a complex interplay between individual and contextual factors that interact diversely throughout the life course. For instance, fertility decline at young ages (30 or under) has been attributed to various social changes since the 1970s. These contextual changes include a lengthening of time spent in higher education (Ní Bhrolcháin and Beaujouan 2012), the rise of effective birth control methods, rising job insecurity, and legalized access to abortion (Bajos et al. 2013; Goldin and Katz 2002). These factors all destabilize traditional gender configurations, opening new possibilities for women. There are also a host of individual factors that have been shown to influence fertility patterns in under-30s, such as ideational shifts, changing gender norms, willingness to cohabitate or be in a couple before considering parenthood (Mazuy 2006), prioritizing one's professional life (Kreyenfeld et al. 2012), and one's moral stance on abortion/contraception (Goldin and Katz 2002). These examples of individual factors underpin the popular demographic framework dubbed the 'Second Demographic Transition Theory' (SDT), in which sub-replacement fertility is thought to result from a shift towards postmodern values of individualism and self-actualization (Zaidi and Morgan 2017). Introduced by demographers Lesthaeghe and van de Kaa (1986), this popular framework claims that individual value orientations are the main determinants of fertility behaviours, intentions, and outcomes. For proponents of the SDT, ideational changes such as increases in individual autonomy, feminism, and secularization, coupled with changes in family formation (such as higher rates of divorce and children born out of wedlock) are essential to understanding demographic change. Changing attitudes towards gender are a critical part of SDT, as the expectations of men and the roles of women have widened to encompass a range of family planning styles and childrearing configurations (such as not being a homeowner before having children). It is important to note that SDT has been critiqued (Coleman 2004; Bailey et al. 2013) for interpreting difference as longitudinal developmental change, that is, viewing certain countries as 'leaders' and others as 'laggards' (Thornton 2005). Despite these critiques, the popularity of this framework speaks to the

importance of considering contextual, institutional, and individual factors together in understanding fertility declines and gaps. This theory also speaks to the importance of gender dynamics, attitudes, and childrearing styles in relation to fertility patterning.

In terms of people older than 30, Beaujouan and Toulemon (2021) state that postponement can be related to unique individual and contextual factors. For instance, first partner unions in Europe are occurring later, and therefore subsequent re-partnering is also occurring at an older age. These new family formation pathways result in a growing population of older couples who wish to have children and are aided in this goal by improvements to healthcare, pregnancy monitoring, and reduced risks of older childbirth (Kotelchuck 2007). Overall, there is strong evidence that fertility decision-making currently depends on a multitude of factors and that the fertility gap (the gap between intention and outcome) itself is an unstable concept that is subject to change throughout the life course. Even though scholars are well-aware of the significance of individual factors in driving fertility patterns (Lesthaeghe 2010), these factors are often considered tricky to demonstrate (Beaujouan and Toulemon 2021). Therefore, while factors such as educational attainment, wealth, and institutional policy play an important role in driving the fertility gap, due diligence must be done in terms of understanding their interconnection with individual factors at play.

There is another strand of research on the fertility gap that is concerned with fertility intentions; that is, how many children people say they desire. This research often claims to complement large-scale quantitative work by addressing individual-level fertility decision-making. These types of studies unpack the social-psychological processes that influence a person's ideal number of children. A dominant framework is the Theory of Planned Behaviour (TPB) (Ajzen 1991), which scholars argue can model the gap between intended and completed reproduction (Klobas 2011; Luo and Mao 2014; Philipov et al. 2015; Williamson and Lawson 2015). TPB is a broad framework from behavioural science, and it posits that behaviour is predicated on intentionality, which is formed by a combination of three factors: attitude towards the behaviour, subjective norms, and perceived 'actual control' over the outcome of the behaviour. In the case of reproduction, advocates of TPB (Klobas 2011; Luo and Mao 2014) claim that fertility intentions are formed through one's attitude towards children (e.g., being 'family-oriented' as opposed to 'career oriented'), subjective cultural ideas about childbearing (e.g., feeling it is a women's duty to have children), and one's actual ability to biologically reproduce. While TPB is fruitful in understanding why people desire children in the first place, it is less useful in analyzing fertility outcomes (Ajzen and Klobas 2013). Researchers have documented that people typically overestimate or underestimate their desired completed family size, and so in general, fertility intentions are not considered a watertight metric for predicting childbearing behavior (Mencarini et al. 2015). In approaching individual-level questions about fertility, it may be fruitful to explore the different, but related question of what people consider to be the most salient obstacles to their reproductive desires. While scholars know that Britons generally want to meet certain life conditions before having children, such as having a career, owning a house, and being married, Beaujouan and Berghammer (2019) have shown that these traditional preconditions for starting or growing a family often go unrealized among people in the UK, sometimes prolonging postponement until it becomes 'too late'. Therefore, shedding light on the types of everyday obstacles that different people face in fulfilling their reproductive goals is important in enriching our understanding of the UK fertility gap.

*1.2. Do Qualitative Methods Add Value to Research on Reproduction?*

Current population research relies almost exclusively on large-scale population surveys, usually as part of large household surveys that include many topics (e.g., the British Household Panel Survey). Although survey data can provide insightful empirical findings to suggest which factors might be important to people in deciding to have children, there are numerous shortcomings of relying exclusively on survey data collection. Survey data does not lend itself perfectly to the elicitation of subjective preferences, such as attitudes,

preferences, desires, and ideals. In addition, survey questions can suffer from ordering effects, where responses to questions can be biased by the question presented previously. Furthermore, many people feel uncertain about their fertility intentions, and often provide answers that are aligned with social norms and post-hoc rationalizations, preferring not to reveal their true opinions (Fledderjohann and Barnes 2018). It is impossible to account for these methodological biases when relying solely on survey data, and therefore, coupling survey data with other forms of data collection is imperative to enrich our knowledge of the fertility gap derived from macro-level research. Ajzen and Klobas (2013) argue that a useful approach in identifying subjective fertility beliefs is qualitative, using an open-ended technique to better identify people's beliefs about family formation. While a qualitative methodology does not completely overcome the methodological issues of survey collection (i.e., respondents can still align their answers were social norms), qualitative data does lend itself well to understanding the motivations of individuals to postpone or refrain from childbearing entirely (Von Der Lippe and Fuhrer 2004; Peddie and Teijlingen 2005). Even though some scholars have elicited information about attitudes from surveys, comprehensive information about opinions, norms, and identity can be difficult to fully understand from questionnaire data alone. Exploring qualitative options can strengthen quantitative findings, providing richness (Peddie and Teijlingen 2005). Recent efforts using mixed methods have been shown to be fruitful in addressing population patterns (Petit et al. 2020).

There have been dozens of efforts to understand fertility from a qualitative perspective, all examining other countries outside the UK (e.g., Taiwan, See: Huang et al. 2022). Notably, Safari-Faramani et al. (2018) explored perceptions of childbearing barriers among a low-fertility subgroup of people in Iran by using a qualitative approach. They interviewed 22 highly educated people in Southeast Iran and determined that there are a host of factors that influenced their participant's fertility desires. This study was comprised of both men and women, all of whom were either faculty members or Ph.D. candidates at a university; therefore, the results of this study speak to the specific experiences of highly educated people in one institutional context. Moving forward, it is integral to consider a wider gradient of educational backgrounds in qualitative research on fertility desires and motivations. There have been several other qualitative studies conducted in this field, predominantly addressing questions of fertility from a health perspective. Specifically, there exists copious research on the role of digital technologies in family planning and reproductive healthcare (Lupton and Maslen 2019; Bailey 2021; Gambier-Ross et al. 2018) and the role that disability and physical health plays in reproductive decision-making (Myring et al. 2011; Sahota and Sankar 2020). Deeper knowledge of the barriers that people face in realizing their ideal family size is necessary to formulate truly targeted policy. Furthermore, since research has shown that reproductive decision-making can be context-dependent (Ajzen and Klobas 2013), a UK-based study is appropriate. It is in response to this literature that this paper contributes a pilot qualitative study to examine the perceived barriers to reproduction among a small sample of men and women in the UK.

## 2. Materials and Methods

We conducted seven focus groups, each with between three and eight participants (with a total of 41 participants), using the online meetings platform, Zoom, during September and October of 2021. Both men and women were recruited for the focus groups through social media (Facebook and Twitter) and snowball sampling. People who expressed interest and fit the selection criteria (male or female of childbearing age, recent parent or planning on having a child, and resident in the UK) were invited to attend a focus group discussion. The criteria of 'planning on having children' was defined through a participant's self-reported active desire to have a child within the next 5 years, which was determined through an online recruitment questionnaire. Participants with more imminent plans were prioritized in terms of focus group invitations. Being a qualitative study, the criterion for inclusion was purposefully broad and non-representative. Since this study did not test an

existing theory or hypothesis, a grounded theory approach was used to produce results. Participants were compensated with a 25 GBP shopping voucher for their involvement in the study.

The goal of the online focus groups was to elicit conversation about individual fertility desires and factors of family planning. Since this study is part of a larger pilot study, one of the main reasons that focus groups were chosen over one-on-one interviewing was that the researchers sought to understand at what point focus groups reach saturation point. In addition, the preliminary and exploratory nature of this project was conducive to the selection of focus groups as a method of data collection, since the authors wanted to elicit broad conversations that cover many topics to design a subsequent portion of the pilot. Respondents were asked a series of questions designed to stimulate diverse discussion surrounding their preconditions for having children. Questions were generally framed through individual experience (e.g., tell me about what you would like to have in place before having a child? What are some things that would prevent you from having a child?). In addition to speaking about themselves, participants were also asked to comment on what they perceive more broadly in the UK (e.g., what are some things that you think make it generally difficult to have children in the UK?).

All the focus groups were conducted by Author 1, who followed a consistent interview schedule (See Appendix A). Two groups included a total of eleven women with university degree qualifications (University Women). Two groups included a total of ten women with vocational, college, or secondary school qualifications (Non-University Women). Two groups included a total of ten men with university degree qualifications (University Men). One group included a total of ten men with vocational, college, or secondary school qualifications (Non-University Men).

To promote openness and more candid conversation, we held the focus groups within the gender-education categories, e.g., university men together. Hinton and Miller (2013) demonstrated in their research on male infertility that men are often stigmatized in reproductive spheres, and thus have a harder time talking about their journeys surrounding fertility and family planning. By interviewing men among themselves, our study sought to create a comfortable digital environment for open dialogue. Despite the attention paid to group cohesion, we also ensured that each group was composed of diverse subjects in terms of variables such as ethnicity, occupation, familial status (whether someone already has children), and geographic location in the UK. Seventeen participants identified as being of White British ethnicity, while the other 24 were British Black, British Asian, White European, Mixed, or another ethnicity. Participants comprised a mix of childless adults and already-parents. The average number of existing children among already-parents was similar from group to group (1.8 overall). The average desired family size among all participants in this study was ~2.5 children (some respondents were uncertain). All participants in this study have been assigned pseudonyms. Table 1 summarizes the participants by group type.

**Table 1.** Participant Summary.

| Group Type | Mean Age in Years (Range) | Proportion Childless | Proportion Partnered |
|---|---|---|---|
| University Women | 30 (24–43) | 64% | 70% |
| Non-University Women | 31 (27–39) | 20% | 90% |
| University Men | 27 (23–32) | 80% | 70% |
| Non-University Men | 32 (25–42) | 40% | 90% |

Since this study is concerned with future family planning, it is important to note that being 'partnered' was defined as currently being with a serious long-term partner. Some participants were not currently with a partner but were formerly with the mother or father of one of their children. In this study, this did not count as being 'partnered', since it was not current.

The focus group audio recordings were transcribed using professional transcription software. The transcripts were then analyzed using a method of inductive thematic coding

described by Wilson (2018) in her qualitative work on reproductive health services. Thematic coding was chosen as the method of data analysis because this method allowed for the systematic categorization of many diverse conversations across different group types. Since the results of this study are wide-ranging, understanding the core themes of the transcripts was essential. This analysis process involved the repeated reading of transcripts to generate a series of codes applicable to all transcripts, which were then grouped to identify general attributes of the text. To conduct the analysis, transcriptions were first divided into sections containing one question and various responses. Second, these sections were inductively coded line-by-line, leading to the pinpointing of 83 initial open codes. Upon the completion of this first round of coding, these 83 codes were organized axially around 22 supra-ordinate categories. The transcripts were then re-coded using these 22 supra-ordinate categories. See Appendix B for a list of the themes, categories, and codes. By counting the occurrences of each category within the re-coded text, six final themes were determined.

People do not yet want children:

1. If they have not secured an external means of childcare or support. (Theme name: Support/Childcare Barriers).
2. If they perceive themselves to be financially or materially insecure. (Theme name: Material Barriers).
3. If they do not feel emotionally mature or stable enough. (Theme name: Emotional Barriers).
4. If they feel that they are not living in an appropriate geographic location for raising children. (Theme name: Spatial Barriers).
5. If they feel that becoming a parent will severely damage their career prospects. (Theme name: Workplace Barriers).
6. If they feel that they are too old to have an uncomplicated pregnancy or delivery. (Theme name: Health/Conception Barriers).

The coding was completed by Author 1. Several verification procedures were followed. First, retrospective re-coding against all transcripts was undertaken numerous times to confirm the applicability and generalizability of each theme across the study. Any new codes generated were constantly checked against the contributions of all participants. This method of verification is described by Nowell et al. (2017) in their article on how to ensure reliability in qualitative data analysis. In our study, some quotes used were edited for grammatical soundness and all identifying information (names of workplaces, schools, etc.) has been obscured from the quotes.

All participants were provided with a study information sheet, and everyone was given an opportunity to ask questions before, during, and after the focus groups. Participants provided their written consent before taking part in this research. The project was approved by the Oxford Central University Research Ethics Committee on 7 August 2021 (Reference SAME_C1A_21-079). Participants were allowed to leave the group at any point without penalty. This research was funded by the John Fell Fund.

## 3. Results

### 3.1. University Women

Groups with university-educated women featured debates surrounding support and childcare, where respondents outlined numerous forms of acceptable childcare—parental support, daycare, support of friends, community support, and partner support. Being unable to secure a source of childcare beyond oneself was generally considered a 'dealbreaker' condition for reproduction by this group.

A common topic in these conversations about support by university women was the idea of partner support. Previous research has established that partner support is a major contributor to maternal mood moderation, with factors such as communication, conflict and instrumental support structuring the maternal experience (Pilkington et al. 2015, 2016). One woman who strongly advocated for the role of partner support in terms of

childcare was Caitlin, a 27-year-old Art Gallery Supervisor from Southwest England. In planning for her first child, Caitlin is actively seeking out, "a partner that wants to be a supportive parent, you know, because, unfortunately, a lot of the responsibility does fall on the mom, especially with breastfeeding". While some participants emphasized the role of love and mutual care, generally partner support was framed in terms of division of practical childcare responsibilities. Government worker Rebecca (30 years old) claimed that after waiting many years to become financially stable, she has now decided to start planning for her first child with her long-term (10 years together) partner, highlighting her partner's role in future childcare responsibility—"It is more important that my partner is on board and understands, I need him to be as supportive as he can. You know, which I hope he will be, and he says he will. We will see how many nappies he's going to change, I suppose". The desire to find a partner who is willing to divide childcare responsibilities equally was widely shared among this group of women and was also a theme among women without degrees. Other forms of support (parental, community, friendships), were also highlighted, demonstrating a strong desire among participants for a holistic support network for raising children. The emphasis on partner support among the university women in the study comprises an important example of 'reproductive support seeking', which researchers have defined as an essential consideration in understanding the lived experience of reproductive decision-making (Clarke et al. 2020). Social support is thought to come in a myriad of forms: informational, emotional, instrumental, affirmational, or appraisal (Baheiraei et al. 2012; Heaney and Israel 2008).

Workplace barriers were another topic of conversation among university women. These women defined workplace barriers as obstacles to reproduction contingent on one's career trajectory. For example, whether a person feels they work in an accommodating company in terms of balancing work and parenthood. The most common type of workplace barrier mentioned by these participants was the notion of career establishment. For instance, mother-of-one, Helen, claimed that becoming secure in her career was her top priority before deciding to have her first child: "And you know for me, having a career is a big thing, you know, so I'm 28 now. And it sort of seemed like the right time, sort of, um, how many years, probably six, seven years into my HR career . . . but yeah, you know, career wise, that was the most important thing, you know, get that out of the way, get myself established". Participants defined career establishment as having enough professional security to be able and take time off for maternity leave without negative consequences to their professional advancement. The primary way that participants claimed they seek to satisfy this condition is either by 'climbing the ladder' in their chosen industry or 'proving their worth' in one solid position for many years before deciding to have children. Emily, a 26-year-old Ph.D. student who hopes to work as an academic, claimed that she would like to be quite far along her career trajectory before deciding to have children—"But if I was heading into becoming a professor or something, then I would not have time to take time off. Otherwise, I would just lose out when a man does not need to take time off because he has a child . . . I want not to be looked down upon because I have a child [at her workplace], which is yeah, difficult. Not many workplaces are like that in the UK . . . in terms of wages and in terms of experience, women are not as highly regarded in the workplace, don't get promotions, don't get any of that. I think it makes it harder for me to have children, knowing it will leave me in a worse position than what I was in before". In this balance between precarious career aspirations and a desire to have children, Emily is planning on having her ideal family of two children, "basically, as late as possible". If Emily does not achieve a sense of job security before it is 'too late', she said that she will forgo having children. New mother of one Eva (35 years old), explained that she also wanted to reach an established point in her career as an architect before having children, but upon realizing that career stability was "not something I could ever rely on happening", Eva decided it was "never going to be the best time to have children, so I decided, OK let's just do it". Unlike Emily, Eva was willing to sacrifice some career progression to have a child. Overall, childless women under 30 reported wanting to wait until they have "accomplished

everything" (Monica, 27 years old) they want in their career before having children, with some participants saying they would ultimately choose their career over family if need be, and other participants saying the opposite. University women over 30 who have children or are planning for their first child report that they waited until their 30s because of career costs, with some citing an eventual trade-off between family life and career advancement to have children before turning 40. Canadian researchers recently indicated that those who have completed a post-graduate degree are three times more likely to report that a lack of workplace support affected their childbearing decisions, typically manifesting as postponement (Metcalfe et al. 2014). Overall, career establishment was emphasized by this group, demonstrating the importance that these university-educated women place on balancing their family and careers, two spheres that are often thought of as incompatible by the public (Sobotka 2004).

Overall, the top two concerns for university-educated women in terms of their family planning were making sure they are with the right partner and making sure that they can balance family life with their career. These two ideas were complemented by an array of secondary barriers to reproduction, such as emotional barriers, material barriers, health/conception barriers, and spatial barriers.

### 3.2. Non-University Women

Like university women, conversations with non-university women surrounded ideas of maternal support, reinforcing a relationality of female reproductive decision-making. In addition to support from one's partner, support from parents was also discussed in these groups. These groups of women defined parental support as the ability and willingness of one's parents to assist with practical childcare tasks such as babysitting. Mother-of-four Amy (35 years old) emphasized the central role that her mother played when Amy planned each of her pregnancies. Without the moral and practical support of her mother, Amy claimed that she would not have chosen to have children. She explained; "It was whether she was prepared to have the baby while I went to work. Whether she'd be at the birth with me because I always want my mom". Mother-of-one Rosa (27 years old) agreed with Amy but explained that the inconsistent support of her parents makes it hard to plan for a second child—"well, first of all for me I always wanted a small family. But I actually felt that after giving birth, I wanted more children. But then looking at how the situation has changed, how high prices have gone, it is very hard to get money these days. And then at the end of the day, children demand so much of your time and with our parents, they started requesting only for certain days . . . I feel as if I am at a point where like, I don't really have enough family support". Amy claimed that when both she and her husband had full-time jobs (Amy is currently a stay-at-home mother), she relied on her mother to care of the children before and after school days.

The importance of having "loved ones stand by" (Rosa) to assist with childcare was linked to another topic of conversation for non-university women—access to formal daycare. The notion of parental support seemed salient for women who could not afford or wished not to pay for, childcare. As Amy explained: "Oh yeah, I couldn't afford to put my children in childcare. Even just breakfast would cost five pounds before school and then five pounds after school. So, if you times that by three kids, it works out to quite a lot . . . (her mom) used to do it before they went to school. But now with my new baby, I've stayed off work". For participants without parents close by or parents that were available to help, enrolling children in daycare "is the only option" (Rosa). Figuring out how to manage the financial burden of daycare was defined as an important factor in family planning. Mother of one Jill (30 years old) claimed that she always wanted to have three children, but that "we thought we knew how much children cost going in, but then they turn out to be a hell of a lot more expensive than that . . . childcare just eats up my whole pay cheque and in terms of planning more, this has really put me off." Jill originally wanted to have three children quite close together so that the children would be similar in age, but now says that having

her first recently has changed her thinking about having three—"Overall financial stuff is the most likely thing to stop me".

Thirty-one-year-old mother-of-five Sandra would like for her parents to help with her children while she and her partner are at work, but her parents suffer from health conditions and are not physically capable of taking care of children. As a result of this, Sandra struggled with making daycare arrangements for her children as she expanded her family in the past. She explained:

> Childcare costs are one of the biggest issues, especially if you are single or if you are in a two-parents family, because you both must go out to work. That is very difficult, because it's very, very, very expensive to put a child into childcare . . . I don't know how we survived it. My children went to nursery, sometimes they went to private nursery, but now there is some sort of funding that helped us, as long as you work, you can put them in school full time when they're not at the school nursery, and it's covered nine till' three. Also, I used to do childcare. So, what I did was kind of double up and take my children to work. That reduced the fees massively, by about 70%. So, these things have come into play for me heavily. Now you get like, sort of 70% from the government because they want us to work so badly, you know, because it benefits Britain and all that. They try and give us all these incentives, but it still doesn't work because they're expecting people to pay childcare upfront. And it's like, how do you expect someone to pay a 1000-pound childcare bill up front when they need a job with childcare to pay the 1000 pounds to pay the childcare? It's a horrible cycle. It's like credit, no one wants to give you credit until you have credit.

This quote from Sandra shows that for those that cannot rely on parental support, paying for daycare is a crucial concern for children with two working parents. Her experience with navigating governmental funding, coupled with her previous career as a childminder, also points to the ways that for families who cannot easily afford childcare, finding solutions might require creativity and/or outside assistance.

These themes of childcare support the previous finding that compared to men, women place significantly more importance on access to childcare in their personal readiness for children (Skoog Svanberg et al. 2006). In a recent large-scale fertility decision-making project, researchers determined significant gender and country differences in terms of how couples report their readiness for children (Boivin et al. 2018). While some factors (e.g., desire to start a family) were universal between countries, other factors such as relational readiness (e.g., access to social support) varied across countries, with women valuing relational readiness higher on average than men. Of the 18 countries analyzed in this large-scale work, the UK and Denmark showed the highest ratings for relational readiness in terms of reproductive decision-making. In their emphasis on varying forms of social support, the results of this qualitative study's focus groups provide richness to previous large-scale quantitative work.

### 3.3. University Men

University men also discussed ideas related to support and childcare frequently, but the prevalence of this theme seemed to be matched by ideas of finance. A common form of support mentioned among these participants was partner support. Respondents emphasized the central role of their female partner in terms of family planning, sometimes deferring questions to what they perceive their female partners to want—"I really think a lot of my answers will come out this way, it really depends on my partner wants" (Dylan). This sentiment was generally framed in terms of female bodily autonomy—"I would like kids soon, but it is not really my choice to make" (Jack, 25 years old). Previous research has shown that men rank 'partner's desire for a child' as a more influential factor in their reproductive decision-making than women (Boivin et al. 2018).

These men championed the prime importance of financial stability and security in terms of their family planning. These participants defined material barriers to reproduction

as any financial precondition for having a child. Examples of these included owning a home, owning a car, having a certain income threshold, and/or long-term income security. These men expressed a dealbreaker attitude in terms of being materially secure before having children. Respondents hoped to be able to afford the costs of things such as education, essentials, holidays, and luxury goods for their children. Father-of-one Tom, 32, was a strong advocate for the importance of financial stability and long-term security. When probed as to what he expected to pay for, he replied—"I don't know, you have to pay for nursery but then you start thinking down the line, okay, well, is going to university important or is it more going into trades? What's important? What if they both want to go to university? Well, are you able to provide for both, when maybe with one, you're able to provide those opportunities? But you also want to keep your own self like, this might sound selfish, but we keep our standard of living a little bit higher, as opposed to having to find ways to survive, which sounds a bit, you know, harsh". While Tom briefly mentioned the cost of childcare, his desire for more wealth was centered more around long-term considerations, that is, what type of life he can provide for his child without sacrificing personal comfort with his salary as a consultant. Tom wants to save more money before having a second child so that he can provide the financial conditions for the long-term success of his children. He claims that he wants to have two children, but that he will not have a second child if he feels he cannot provide financially. In general, participants expressed a strong sense of financial responsibility in terms of deciding to have a child. Ph.D. student Lucas (24 years old) claimed that he would like a steady job and a better pay cheque before having a first child because he feels an important "sense of personal responsibility, to put the kids before myself". Despite reporting a very strong desire to have children (Lucas is currently childless)—"Oh my desire for children is like a 10/10", Lucas insisted that feeling financially insecure would be a deal-breaker for him in terms of having children. In their questionnaire study on male reproductive decision-making in Sweden, Bodin et al. (2019) determined that among men who report a visceral desire for children, their reasons for remaining childless almost always concern practical reasons. The emphasis that these select male participants placed on economic issues reinforces the dealbreaker nature of finances in male decision-making.

As mentioned above, some university-educated women claimed that they wanted to postpone their family planning into their 30s, primarily to advance their careers. Some university-educated male participants also reported the same desire to have children in their 30s, but for a superficially different reason—to capitalize on the perceived independence of being childless in their 20s. This sentiment was thematically coded under the general theme 'emotional barriers'. For instance, 24-year-old fashion industry worker Lewis explained, "I would of course like to be at a stage in my life where I'm financially stable enough to have kids, which I am not at the moment. But the thing that overrides everything in terms of waiting is me wanting to live my life on my own for a while longer before having children. And for me, I would say that that would be earliest, like 30 to 33 years old. I want to have my 20s childless for sure. Wanting to live a little is more of a justification for me than the financial justification, which is though of course, still important". In response to this comment, Tom replied, "I really share Lewis's viewpoint. So, I really didn't want kids in my 20s. I worked so hard to go through uni and to do all that, I just wanted some time to myself. The big thing for me was financial security, so not having children until I was in a stable enough position to provide for them. So, it has taken me until 31/32 years old, both as a mix of selfishness, wanting to travel and this sort of thing, and also saving money". For financially secure 28-year-old childless software engineer Aaron, his desire to wait until having children with his partner is almost entirely rooted in notions of personal freedom—"I want to go out and do a lot of adventures and activities, like skiing, vacations, and so on. I expect that with kids, all of this will become a lot harder to organize because you have to take kids into account. I expect there will come a point quite soon where I'll be like more settled and my life will be routine. Until I'm there, I want to wait a bit more". The perceived impact of children on one's personal freedom was a major talking point among

men in these focus groups, with some citing it as a main factor in their reproductive decision-making. When probed about why these men thought children impact their freedom, Aaron clarified that "children just need a lot of time, attention, and care. Someone needs to be there to resolve things and that means I need to use my free time to manage kids". For Aaron, he expects to play a significant role in childcare, and thus wants to wait until he has had 'adventures' before dedicating his free time to childcare. For most of the other participants, the conversation around personal freedom was more centered around their leisure time and mobility. As 31-year-old data scientist George explained—"So the other time we (him and his partner) wanted a family we decided we were not mentally prepared enough because we were too young. We had not seen the world. We had not spent much time together as partners. Like traveling, things like that". This explanation relates strongly to the idea of career establishment among women, that is, wanting to accomplish goals in one's 20s before becoming encumbered by children. Researchers claim that 'personal development and self-image' is a prominent self-reported reason that men ultimately choose to plan for children (Bodin et al. 2019), further emphasizing the notion that subjective norms can play a significant role in structuring male fertility desires.

Overall, in deciding when to have children, this small select group of university-educated men prioritized the desires of their partners, their perceived material wealth, and emotional readiness for children.

*3.4. Non-University Men*

Conversations with non-university-educated male participants favoured financial themes. Almost all of these participants expressed a desire for increased wealth, but for a variety of different reasons. For instance, Samuel, 33, claimed that finances are his top consideration in family planning because he values being able to pay for things "like, clothing for example, because we know that after the first two months, like a kid will like outgrow the clothes so you need to do like regular shopping, we need to like do regular shops for everything". Richard explained that he is saving money so that he and his wife can have an adequate emergency fund for their child. Richard also wishes to be able to pay for holidays and picnics for his entire extended family. Some respondents insisted that they desired financial gain to pay off debts and loans "being not in debt and having a few pounds saved up from work or investment would be ideal" (Kabir). For 41-year-old father-of-one Ibrahim, saving money is integral so that he can ensure a harmonious home life with his wife, who he tries to avoid "haggling about money with, and then the family would be unstable". These rationales for saving money were generally short-term in nature, with participants emphasizing the everyday expenses of having a baby. One notable participant named Glen (25 years old) claimed that he and his partner want to have eight children and that they will start trying next year. For Glen, he claims that income itself is not a huge issue (Glen works a stable job in the military), but that he needs to be able to afford a house big enough for eight children. When probed as to whether the size of his house is a dealbreaker, he claimed: "We would pause before having another one if we didn't have the space". For Glen, the cost of childcare is not a relevant barrier to his desired family size since his wife wants to be a stay-at-home mom. He explained that the only insurmountable barriers that would keep them from having a very large family would be space in the house and space in the car— having a large car was of prime importance to Glen.

Interestingly, considerations of one's living environment were of importance to non-university men in this study, which was thematically coded as 'spatial barriers'. A spatial barrier can be defined as any obstacle to family planning that involves one's geographic location in the UK. An example of a prominent spatial barrier discussed in focus groups is a person's perception of the appropriateness of one's neighbourhood for raising children. For example, father-of-one Ibrahim claimed he would like to live in a quieter area before having another child. He was particularly adamant about not living in a high rise or living close to the road, claiming that these types of living situations are not safe for children. Ibrahim

also emphasized that when he lived in London (he now lives in the Greater Manchester Area), he postponed having children because of his negative perception of the hectic urban environment—"There were like, a lot of kids were going into gangs, and they were having a bad influence on the other children around. So that bad influence would always lead children astray. It will lead them into being gangsters, you know, using selling drugs on the street and all this kind of stuff. So, the area that I was living in then was discouraging me at one stage from having children. We wanted a good environment to have children". Even though he does not live in London anymore, Ibrahim still does not perceive his geographic location to be child friendly and wishes to move again before having his next child. Ibrahim claimed that these environmental considerations were his top deal breaker in terms of having more children and that he would not have a second child (something that he has always wanted) unless he moves. Martin, a 33-year-old delivery driver who lives in Milton Keynes agreed wholeheartedly with Ibrahim—"I know exactly what Ibrahim is talking about. That greenery is essential, man . . . . the pressures from society as well, you know the markets and everything, there is a lot of uncertainty about how the world is rolling. I want to make sure when I bring another child into my environment, it has to feel more like a place to thrive in than to just exist in, you know". By emphasizing how his choice of neighbourhood connects to broader fears he has about UK society, Martin shows that one's perception of spatial barriers can connect to more general emotional barriers and concerns for the future. Interestingly, Martin made an explicit distinction in terms of his individual barriers to reproduction. He defined some of his barriers as "you know, regular stuff. Like childcare, if food is going out, if I have a car" and other barriers such as "everything else, like pressure from society and where I live". This distinction that Martin makes between practical barriers to reproduction and less tangible barriers to reproduction summarized well the conversations in all focus groups, which would fluctuate between practical and non-practical concerns. Overall, some interesting ideas that emerged from non-university male participants were the importance of short-term material wealth and the importance of living in a child-friendly location.

## 4. Discussion

The results of this qualitative study can inform scholars on some of the possible barriers to reproduction among British men and women. Knowing that fertility patterns differ across educational strata in the UK, focus groups were carried out separately for university-educated and non-university participants. In summary, university women emphasized the role of their careers and partner support in terms of family planning, whereas non-university women discussed parental support and paid childcare options. University men valued financial stability and personal readiness for parenting, whereas short-term material wealth and the importance of living in a child-friendly location were discussed among non-university male participants. In general, the focus group discussions across group types featured many complex themes that often overlapped in significant ways.

First, focus group conversations pointed to the importance of support and childcare in terms of how people make decisions about their reproduction. In addition, the type of support that is valued the most varied, with some participants emphasizing support from one's partner and other participants emphasizing support from one's parents. It is important to note that the term 'partner support' was explicitly used (with some variations, e.g., 'the support of my partner') by female participants. Just having a partner was not enough for female participants, who claimed that their partners need to be both practically and emotionally present throughout childrearing. It is interesting, though, that the term 'partner support' implicitly suggests an unequal division of childrearing labour (with one partner as the 'preferred parent' and the other as support), though female respondents explicitly claimed to prefer more egalitarian parenting. This could possibly indicate that female participants desired more egalitarian home situations, but that the male-breadwinner-female-homemaker model still dominates the discourses and realities of parenting. The male-breadwinner-female-homemaker model is a dominant cultural

model that espouses household role specialization, with women as homemakers and men as breadwinners (Raybould and Sear 2021). Our findings support previous research that determined traditional gender roles still proliferate in the UK, and that the egalitarian caregiver model has a utopian character (Ciccia and Bleijenbergh 2014). Overall, the hierarchal nature of the term 'partner support' coupled with reported desires for egalitarian home configurations illuminates the complex relationship between women's realities of parenting vs. their idealized versions.

The emphasis that women place on childcare supports Schaffnit and Sear's (2017) finding that the availability of childcare is an important factor that determines reproductive decision-making in the UK. They also support Safari-Faramani et al. (2018) qualitative findings from Iran that claim 'problems with childcare' as a primary category in their analysis. These results can also be used to enrich the previous quantitative finding that the specific type of childcare support that is valued by someone is influenced by socioeconomic conditions (Schaffnit and Sear 2017). In addition, Van Bavel and Klesment (2017) writes that with the rise of female education and gender education gap reversal, working women are more likely to prioritize partnering with a man who provides childcare and does household chores but, as mentioned above, this might sometimes represent an ideal rather than a reality. With support and childcare emerging as a prominent theme in this study's analysis, these ideas warrant a more dedicated investigation into understanding the fertility gap. These focus group results also affirm the previous finding that social support is valued by women, who scholars claim experience their health more relationally than men (Birditt and Newton 2017; Clarke et al. 2020; Haines et al. 2008). In creating childcare policies to address the fertility gap, it is important to understand the different forms of childcare and support (partner, parental, community, etc.), considering intersections of education and wealth. Since highly educated women in the UK have the highest rates of excess childlessness (Beaujouan and Berghammer 2019), improving policies that aid work-family reconciliation is of paramount importance.

Another interesting finding of this study is that considerations of opportunity cost and workplace opportunity are defined as insurmountable barriers mostly among university-educated female participants in focus groups. This notion supports previous research that has found that career advancement is a major contributor to postponement, but that this is mainly considered crucial to those with university degrees. In addition, Safari-Faramani et al. (2018) defined the 'incompatibility of work and family for women' as one of the main self-perceived barriers to reproduction among highly educated Iranian women in their qualitative study, demonstrating a cross-cultural commonality between different qualitative studies in this subject. With there being extensive literature about female education and fertility postponement, our study provides some examples of personal rationales as to why and how education might influence reproductive decision-making; this study is a small glimpse into how people themselves view their education, workplace, and family desires. Of particular interest are those people who report postponing childbearing for reasons of career establishment but are working in precarious industries (such as academia). For instance, academic research careers are commonly marked by fixed-term positions that lack continuous employment prospects (OECD 2021). The reported reasons for these select women's desires to postpone are not financial, but rather reflecting a possible factor of social capital in driving career desires. These results demonstrate the potential influence of social capital in structuring fertility desires and timing, which could be obtained through education (Philipov et al. 2006). Overall, focus group conversations among university-educated women pointed to the methodological importance of bottom-up approaches that capture the difference in subjective values between those who are university-educated and those who are not. By further exploring the role that career establishment can play in terms of education level, we can have a more holistic understanding of different values and life trajectories along lines of socioeconomic positionality.

Finally, male participants in this pilot study suggested that financial considerations were of prime importance to them. Particularly, short-term material barriers were discussed

in focus groups with non-university educated men and being able to maintain a certain standard of living was discussed among university men. These results illuminate some gendered attitudes towards the male-breadwinner-female-homemaker model (Raybould and Sear 2021). It seems that the male participants in this study structured their fertility desires through their ability to fulfill their roles as breadwinners while balancing their desires for personal freedom. Similarly, the emphasis that female participants in this study placed on childcare may be explained by cultural pressures to be the main homemaker. It was interesting, however, that female participants did not prioritize the material wealth of their male partners in terms of their partner selection. This counters the male-breadwinner-female-homemaker model, in which women seek out male mates based on one's ability to provide financially. Rather, the importance female participants bestowed upon egalitarian homemaking and childcare speaks to how gendered ideals about divisions of labour can influence fertility desires. Raybould and Sear (2021) claim that working women experience a dual burden of balancing paid work with childrearing, leading to lower fertility desires. They also explain that with men being culturally expected to take on the entire financial burden of childrearing, this can result in men wanting fewer children. In their qualitative research in Spain, Bueno and Brinton (2019) found that more egalitarian couples were more resistant to economic instability and maintained more stable fertility desires. With the female participants in this study seeming to prioritize having a more egalitarian partner in terms of childrearing, but the male participants strongly prioritizing their ability to provide financially, one can see observe a possible tension between 'what women want' and 'what men want', owing to dynamic gender roles and distinctions. This supports the central tenets of the SDT, which espouses a breakdown of hegemonic models of childbearing, financial provision, marriage, and childrearing.

Finally, these findings constitute our most up-to-date qualitative information on what men want in place before having children. To date, few studies have analyzed the role that men play in reproductive decision-making, and most of this literature considers the couple dynamic because men's fertility has been strongly statistically linked to their selection of similarly socially situated female mates (Trimarchi and Van Bavel 2017). Men included in reproductive decision-making research are usually thought of as one-half of a heterosexual couple or as men who are voluntarily childless (Bodin et al. 2019). By asking men themselves what they want around family planning, this study constitutes an example of what men might potentially want when asked about it without their partners present.

This study is constrained by several meaningful limitations. It is important to account for the differences in mean age and childlessness between group types since this could have influenced the most important barriers to reproduction identified in the thematic analysis. In addition, due to the size of this study, the results have not been explicitly connected to partnership status—an important determinant of fertility desires. For this reason, it is not possible to meaningfully compare any of the themes discussed between groups, and the results reflect the non-generalizable nature of this study. By mixing parents with childless participants, we recognize that the results of this study are informed by a diverse range of background experiences; however, we do not believe that the opinions of parents or non-parents were compromised by being in mixed groups. Being an exploratory pilot study, we plan to expand this research to other contexts, where we can hold separate discussions for those wishing to become first-time parents and those wanting to expand their families to see if this changes the discussion dynamic.

Quantitative work has shown that already having a child significantly alters fertility intentions and barriers since families often learn some of the realities of child-rearing (Ajzen and Klobas 2013). In addition, with some subjects reporting intersectional identities (being foreign-born, having a mixed cultural background, being a part of the LGBTQ+ community), our results are certainly influenced by the specific combination of background factors present, as in all qualitative research, that we could not consider. We do not feel that these contextual factors detract from our findings, since they are meant to merely illuminate some potential barriers to reproduction among the specific participants in this

study. Furthermore, since these focus groups were conducted during the autumn of 2021, the COVID-19 pandemic could have played a role in influencing some respondents. Ample research has been conducted during the COVID-19 pandemic to determine its influence on reproductive desires and experiences (Gray and Barnett 2022; Wright 2022). These studies have shown that the insecurity, fear, and limited social interaction associated with the COVID-19 pandemic have had a significant impact on family planning (Wright 2022). In this study, participants did not explicitly mention the pandemic prominently, though there may have been an implicit influence structuring their responses. Finally, individual accounts of reproductive decision-making cannot necessarily be drawn out from focus groups, since group discussions can sometimes be influenced by inter-participant dynamics and social pressure. For instance, the emphasis that university women placed on career advancement might have potentially been influenced by dynamics of social desirability. Due to the possible pressure of gender roles that frame female desires for personal freedom as selfish (Cooney 2020), one can speculate that some female participants may have cited their careers as main barriers to reproduction to appear more socially acceptable in this highly educated group environment. Knowing that the other participants in the focus group are university educated (many mentioned their experiences in university and work), these female participants may have felt that emphasizing their career opportunity cost was a 'safe' response that other participants would agree with. In combating the possible social desirability limitations of group environments, further research will include one-on-one interviewing to draw out first-person testimony unencumbered by social norms and pressures.

The results of this qualitative study can serve as an important basis for the development of further research to generate robust wide-scale data on the fertility gap. These results also represent an effort to build upon the work of other scholars outside the UK who have used qualitative methodology to understand questions of reproduction, population, and fertility (Bailey 2021; Huang et al. 2022; Safari-Faramani et al. 2018). In addition, these results can inform policies that support the human right to reproduction and family. Further research on identifying the barriers to reproduction can help accumulate the necessary knowledge to start lessening burdens on people who want to have more children.

**Author Contributions:** Conceptualization, P.S.; formal analysis, M.B.; funding acquisition, P.S.; investigation, M.B.; methodology, M.B. and P.S.; validation, M.B.; writing—original draft, M.B. and P.S.; writing—review & editing, M.B. All authors have read and agreed to the published version of the manuscript.

**Funding:** This research was funded by the John Fell Research Fund, grant number 0010698. The John Fell Fund is funded by the Oxford University Press.

**Institutional Review Board Statement:** The study was conducted in accordance with the Declaration of Helsinki, and approved by the Institutional Review Board (or Ethics Committee) of the University of Oxford (Reference: SAME_C1A_21-079, 17 August 2021).

**Informed Consent Statement:** Informed consent was obtained from all subjects involved in the study.

**Data Availability Statement:** Themes and coding are available in Appendices A and B. Raw data are subject to GDPR constraints, but anonymized portions can be made available after the duration of the project is completed.

**Conflicts of Interest:** The authors declare no conflict of interest. The funders had no role in the design of the study; in the collection, analyses, or interpretation of data; in the writing of the manuscript, or in the decision to publish the results.

## Appendix A

Appendix A includes the interview schedule and questionnaire.

| Interview Schedule |
| --- |
| *Introduction/Welcome* |

- Thank participants for their participation in the focus group
- The purpose of the focus group is to learn more about their perspectives on having children for a research project on fertility in the UK—the results will be used to design a bigger experiment in the boarder population
- Will be recording today's focus group—before the recording starts, cover strategies to protect participants' confidentiality

  ○ Eliminating personally identifying information like names, job titles, city names, etc.
  ○ Will use quotes, but never identify the speaker
  ○ No one else except me will have access to today's recording or any transcript from the recording

- Ground rules

  ○ Put up either a digital or real hand to indicate that you have something to contribute

- Questions?

| *Introductory Questions* |
| --- |

1. First, let us go around the digital room for introductions—tell us who you are, how old you are, and if you currently have children or not.
2. Please share with us your ideal future plans in terms of having a family.

   a. How many children do you want?
   b. How many children do you currently have?

3. How old would you like to be before having your first child? OR How old were you when you had your first child?

| *Core questions* |
| --- |

1. What is the number one thing that influences your decision-making surrounding having children?
2. What are some essential things that you would like to have in place before trying to have a(nother) child?

   a. Which of these essential things might you consider to be deal-breakers in terms of trying for children

3. Are there any main factors or things that would prevent you from having kids? OR is there anything that would have prevented you from having kids?
4. What are you some things that you think make it difficult to have children in the UK?
5. How do you plan on supporting your expanding family?
6. How strong is your desire for children?

   a. How important is having children to you?

7. What are some things that you do not consider essential in your plans for children? i.e., things that are 'overlookable', (such as perhaps having a partner or house or something in this vein) factors that you are indifferent about.
8. Anything else?
9. Consolidate discussion—so is it fair to say that you would rate x, y, and z, as the most crucial things that influence your thinking towards children?

| *Closing/Wrap-Up* |
| --- |

- Questions?
- Thank everyone for their participation
- Remind them of their anonymity
- Ask them for feedback
- Discuss incentive distribution
- Ask for follow ups

## Appendix B

Appendix B includes the coding key for the study.

## Coding Key

| Theme | Category | Code |
|---|---|---|
| **Material Barriers** | Home ownership | Having enough space<br>Creating a sense of stability<br>Owning a home<br>Having a big home |
| | Income | Financial Stability<br>Well-paying job<br>Good Salary<br>Financial security<br>Marrying rich<br>Passive Income |
| | Car ownership | Big car<br>Owning a car<br>Family friendly car |
| | Government benefits | Child benefits<br>Stipends<br>Low-income support<br>Employment Support |
| **Emotional Barriers** | Visceral Desire for children | Varying attitudes towards babies<br>Baby Fever<br>Abstract notions of 'wanting one' |
| | Mental Readiness for children | Coping with mental toll of previous child<br>Feelings of personal readiness for children<br>Stable mind/heart<br>Ready to take next step<br>Done things in personal life (i.e., travel)<br>Attending to one's mental health |
| | Perception of the outside world | Worries about the quality of child's future<br>Attitudes towards marginalized groups<br>Fears of climate change<br>COVID-19/pandemic related anxiety |
| | Cultural Pressure/Influence | Desire to conform to typical family size norms<br>Cultural norms when to have children<br>Pressure to have or not have children<br>Conforming to parental wishes |
| **Support/Childcare Barriers** | Support system—friends | Living near friends<br>Friendship Support system<br>Moral support from friends<br>Friends taking care of children<br>Timing children to match friends |
| | Support system—parents/family | Living close to/far away from parents<br>Parents able to do childcare<br>Parents willing to help raise children |
| | Support system—'strangers' | Having friendly neighbours<br>Being surrounded by other parents<br>Mommy networks on/offline<br>Daddy networks on/offline |
| | Having the right partner | Supportive and loving partner<br>Partner is a good mother/father<br>Having a partner close by/living together<br>Compatible parenting styles<br>Being married<br>Partner willing to help with children |
| | Daycare | Cost of nursery and daycare<br>Being close to a good daycare<br>Timing children in line with daycare options<br>Free daycare hours |

## Coding Key

| Theme | Category | Code |
|---|---|---|
| **Health/Conception Barriers** | Medical Support | The role of the NHS<br>Midwives/quality of care<br>Access to information surrounding pregnancy<br>Pre-natal, antenatal, and postnatal support<br>Feelings of being looked after |
| | Perception of Age-Related Issues | Timing children in line with biological clock<br>Fears of personal health complications due to age<br>Fears of the health of baby due to age<br>Having a youthful energy level<br>Feeling 'too old' |
| | Physical Ability to Conceive | Needing to rely on IVF<br>Having inherent fertility Issues<br>Struggling to get pregnant |
| | | Added difficulty of LGBT relationships/IVF |
| | Healthy Birth | Fear of passing down genetic issues<br>Fears of passing on familial diseases<br>Fears of mutations<br>Fears of the baby having a disability |
| **Spatial Barriers** | Support system—parents/family | Living close to/far away from parents<br>Parents able to do childcare physically<br>Parents willing to help raise children in-person |
| | Spatial Considerations | Being settled in one place physically<br>Having a permanent location<br>Living in a child friendly town/place<br>Wanting to raise child in a good environment<br>Living in an undesirable location to raise children |
| | Being Settled down/Lifestyle | Having a stable lifestyle<br>Having a daily routine<br>Being more certain of day-to-day activities |
| **Workplace Barriers** | The Opportunity Cost | Not wanting to compromise an established career<br>Using' a university degree<br>Cannot sacrifice current schooling<br>Accomplishing main professional goals<br>Having the clout to take time off<br>Not wanting to lose out to men professionally |
| | The role of workplace benefits | Having good maternity leave options<br>Possibilities for part-time work<br>Flexible working options<br>Having childcare options at work<br>Secure work contract in terms of good benefits |
| | The role of workplace culture | Having a good work/life balance<br>Accepting work culture of motherhood/fatherhood<br>Working somewhere with other parent |

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
