# Peer review of "Fertility Decision-Making in the UK: Insights from a Qualitative Study among British Men and Women"

_socsci, doi:10.3390/socsci11090409_

Round 1

Reviewer 1 Report (New Reviewer)

Please see the attached document with comments and suggestions

Author Response

Thank you!

Reviewer 2 Report (New Reviewer)

This paper provides interesting qualitative evidence and depth to the topic of fertility, often approached with quantitative methodology. The evidence on men’s views and experiences in family/fertility intentions and planning make a particularly useful contribution to the literature. I am not a trained qualitative researcher so cannot comment on the methods other than to say that as a non-expert I found the ‘Materials and Methods’ section well-explained. I hope the authors nonetheless find my comments useful and constructive. I have two main comments where I think the paper can be improved:  the framing and the discussion of results.

 1)      Framing/focus

The abstract and introduction frame this paper as research into the 'fertility gap' and the six analysis themes identified are listed as “barriers”. Yet the results and discussion do not directly discuss whether participants have or expect to have a completed family size smaller than their ideal family size - and their reasons for this. This is despite the interview guide including explicit questions on the desired and current number of children. Instead, the qualitative evidence seems to mainly revolve around negotiating timing of rather than barriers to having a first or subsequent child. From the results provided it seems that despite some delays and/or difficulties the participants may well anticipate eventually having the number of children they desire.

Although the qualitative evidence presented on the gendered and classed experiences and rationales is interesting and complements existing literature, I am not convinced that in its current form it is as novel and makes as strong a contribution as the paper might do if it directly provided evidence on (unsurmountable) barriers to fulfilling fertility intentions/desires.

2)   Analysis/discussion of results

Given the discussion in the introduction of social desirability influencing participants' responses, I at times wondering if researcher probed for elaboration/clarification. There were instances where I felt the description of the results did not quite deliver on the argument that the researchers had set out of providing greater depth and nuance than can be inferred from the existing (quantitative) literature. For example:

·  The gendered barriers among the highly educated: women citing career establishment as pre-condition to avoid penalty while men cited attaining financial security & prioritizing independence in 20s. Are these men and women really talking about the same thing but using different terminology? If it is less socially acceptable for women to say they want to put themselves first, travel, go out etc., this would not be given spontaneously as a reason for postponement, but waiting for career establishment would also serve that purpose. 

·  Similarly, was there any follow-up from the researcher about the men saying the timing of family formation ultimately being their partner's decision. I found myself wondering if the man quoted meant this specifically in relation to pregnancy her bodily autonomy or (also) a more general expectation that the final say rests with her because she would do the bulk of the caring, adjust her employment pattern etc.

· Was there any follow-up to the expression of remaining childfree in 20s to preserve independence - as to how these men saw their role as fathers. Is this a reflection of the partner support barrier mentioned by women? Are the men (expecting to be) supporters of a main carer or equal carers?

The discussion of 'support' would benefit from more critical reflection and elaboration. A) It was unclear to me whether 'partner support' was the participants' terminology or the authors' interpretation. 'Partner support' suggests an unequal division of parenting (ie a main/supporter hierarchy) yet there was also mention of some highly educated women expecting egalitarian division of childrearing. B) At the moment 'support' is referred to both in relation to partners - i.e. the other (prospective) parent - and extended family (generally maternal grandparents it seems), thus conflating paternal parenting and non-parental childcare which seems primarily to be in relation to enabling (maternal/dual) paid work. 

Other comments:

Page 15; line 583: “In creating policy to address the fertility gap, it is important to address the many forms of childcare, considering intersections of education and wealth.” What about other policies? For example to encourage active paternal involvement in parenting/ egalitarian division of paid and family work (lines 300-301)? Affordability of formal childcare (line 401)? Sufficiency of benefits aimed at families (lines 514-516)? 

Page 16; lines 619-620: It seemed to me that university educated men likewise saw themselves primarily as breadwinners but expressed it in terms of needing to secure/maintain the standard of living they/their family was accustomed to when considering having a(nother) child. 

Author Response

Thanks!

Round 2

Reviewer 2 Report (New Reviewer)

The authors have taken on board suggestions and the revised manuscript more directly addresses the fertility gap and insurmountable barriers. I have no further comments.

This manuscript is a resubmission of an earlier submission. The following is a list of the peer review reports and author responses from that submission.

Round 1

Reviewer 1 Report

The study uses seven focus groups from the UK in order to study the fertility gap. It suffers from a number of drawbacks which need to be addressed before the study can be considered for publication in a scientific journal. The most important ones are: (a) The study quantifies qualitative results; (b) while it cites a lot of important literature, it does not present the current state of the art in a concise way, state the research gap and explain how their study contributes to it; (c) there is no chapter specifically dedicated to previous research (and possibly theory).

Abstract:

The abstract should acknowledge better that a lot of scientific work has addressed the fertility gap (especially the first sentence doesn’t). Qualitative results are quantified (see below).

Introduction:

p.1

The second sentence mixes up lower fertility (which could be a lower number of children) and childlessness. The third sentence should give an indication on the cohorts studied, it reads as if the study refers to current behaviour. Fifth sentence: There are many factors responsible for involuntary childlessness. In some countries, women postpone to relatively high ages, but still catch up quickly.

p. 2

Second paragraph (from line 41): The authors should clarify better in which direction the variables work, e.g. women’s higher education is related to lower fertility and higher childlessness. That would be much more informative.

Third paragraph (from line 53): Some of the arguments are flawed, e.g. of course effects are context-dependent – this only means that policy and institutional characteristics matter for people’s choices. It reads as though it was a drawback of quantitative research. Quantitative studies go far beyond correlational results, there are many studies using longitudinal data that are studying causes. I don’t think the authors need to downplay the important results from quantitative research in order to argue in favour of using qualitative methods.

p. 3

Second paragraph (from line 99): Again, the arguments are not well developed, e.g. there is a long tradition – and a lot of accompanying methodological research – of studying attitudes with survey data. Moreover, some of the arguments presented will also apply to qualitative data (question ordering in more structured qualitative interviews, social desirability bias à this may be even more of an issue in focus group discussions).

p. 4-5

Materials and methods: The authors need to explain why the chose focus groups and not, for instance, qualitative interviews. Focus groups are especially useful when one aims to study societal discourses. Is this the aim of the study? Why did they use thematic coding? The methods’ choices are not well explained.

p. 6

I think that the authors do not really understand the merits and limits of qualitative data. Since their data are not representative, they cannot gain quantitative insights from them – yet, their manuscript is full of quantitative statements. Already in the abstract, we read which groups values what most highly, is most concerned etc. In the discussion of results, we read about most-often mentioned, second-most prevalent, more important to men than women etc. This kind of inference is not possible with qualitative data because it is not representative – if it was, we wouldn’t need expensive, representative surveys. Qualitative data can lead to very important insights e.g. detect novel trends that do unnoticed in quantitative research, inform us on a wide range of possible factors – but it cannot generalize them in a quantitative way. It is also doubtful whether results can be clearly delineated between the four groups. In essence, the authors are establishing a correlation – e.g. university-educated women discuss the issue in a certain way – but based on non-representative data. It could be just this group of a dozen women (selected in unknown ways) who discuss the issue in a certain way. The study reads as though the authors generalize the results of around one dozen women to the many millions of highly-educated women in the UK.

p. 14

Discussion: because the authors have not worked out well the state-of-the-art, it remains unclear how their study contributes to it.

Reviewer 2 Report

This paper aims to provide new qualitative information about fertility intentions in the UK, drawing on focus groups designated by educational status and gender. The authors identify key factors that promote and deter plans and intentions for childbearing.

While I appreciate the topic area, I have concerns about the approach. As the authors note in the Discussion section, ideas about having children are notably affected by whether or not one has children already. Their focus groups range from only 20% who have had children (University men) to fully 80% who have had children (non-University women). It just doesn't seem comparable to ask these questions across these very different family formation contexts. Also, the University Men are notably younger (mean age of 27) than the other groups (mean ages of 30-32). In addition, intentions about future childbearing are seriously affected by partner status, and they are not considering whether individuals have partners/spouses in their designation of groups.

Overall, I am just not sure this paper makes a strong contribution toward understanding fertility intentions given the research design.

Reviewer 3 Report

The article is well written and offers an original contribution to the field despite the limitations in the research design that are described.
My suggestions for the authors are:
1. For the structure:
- divide the Introduction section from the Theory section
- Divide the Data and method section from the Analytical strategy section
2. For the results: it is not clear to what extent the spatial barrier is a 'subjective' characteristic of this group of respondents or an 'objective' condition. The authors could discuss a bit more about the living conditions of the non-university men, e.g. do you have information about their neighbourhood? Do you think these findings are due to their cognitive patterns or the living context that is different from university men?